# Discovery of Carbonic Anhydrase 9 as a Novel CLEC2 Ligand in a Cellular Interactome Screen

**DOI:** 10.3390/cells13242083

**Published:** 2024-12-17

**Authors:** Sebastian Hoffmann, Benedict-Tilman Berger, Liane Rosalie Lucas, Felix Schiele, John Edward Park

**Affiliations:** 1Division of Cancer Immunology and Immune Modulation, Boehringer Ingelheim Pharma GmbH & Co. KG, 88397 Biberach, Germanylianelucas33@gmail.com (L.R.L.); 2Division of High-Throughput Biology, Boehringer Ingelheim Pharma GmbH & Co. KG, 88397 Biberach, Germany; benedict-tilman.berger@boehringer-ingelheim.com (B.-T.B.); felix.schiele@boehringer-ingelheim.com (F.S.); 3Division of Biotherapeutics Discovery, Boehringer Ingelheim Pharma GmbH & Co. KG, 88397 Biberach, Germany

**Keywords:** carbonic anhydrase 9 (CA9, CAH9, CAIX), C-type lectin-like receptor 2 (CLEC2), podoplanin (PDPN), recombinant membrane protein library, platelet-tumor cell aggregation, AlphaLISA screen, monocyte/platelet membrane protein interactome, Hodgkin’s lymphoma, blood cancer, deorphanization, protein-protein interactions, receptor screen, SYK, cell-cell interactions

## Abstract

Membrane proteins, especially extracellular domains, are key therapeutic targets due to their role in cell communication and associations. Yet, their functions and interactions often remain unclear. This study presents a general method to discover interactions of membrane proteins with immune cells and subsequently to deorphanize their respective receptors. We developed a comprehensive recombinant protein library of extracellular domains of human transmembrane proteins and proteins found in the ER-Golgi-lysosomal systems. Using this library, we conducted a flow-cytometric screen that identified several cell surface binding events, including an interaction between carbonic anhydrase 9 (CAH9/CA9/CAIX) and CD14^high^ cells. Further analysis revealed this interaction was indirect and mediated via platelets bound to the monocytes. CA9, best known for its diverse roles in cancer, is a promising therapeutic target. We utilized our library to develop an AlphaLISA high-throughput screening assay, identifying CLEC2 as one robust CA9 binding partner. A five-amino-acid sequence (EDLPT) in CA9, identical to a CLEC2 binding domain in Podoplanin (PDPN), was found to be essential for this interaction. Like PDPN, CA9-induced CLEC2 signaling is mediated via Syk. A Hodgkin’s lymphoma cell line (HDLM-2) endogenously expressing CA9 can activate Syk-dependent CLEC2 signaling, providing enticing evidence for a novel function of CA9 in hematological cancers. In conclusion, we identified numerous interactions with monocytes and platelets and validated one, CA9, as an endogenous CLEC2 ligand. We provide a new list of other putative CA9 interaction partners and uncovered CA9-induced CLEC2 activation, providing new insights for CA9-based therapeutic strategies.

## 1. Introduction

Membrane proteins play a pivotal role in the function of peripheral blood cells, acting as gatekeepers and communicators between the internal cellular environment and the external milieu. This includes the transmission of extracellular cues across the membrane, mediating cell-to-cell or cell-to-matrix attachments, the regulation of ion exchange, and enzymatic activities [1].

Studying membrane protein interactions is essential for understanding their functions but can be challenging due to complex post-translational modifications that are not easily replicated in commonly used expression systems, weak binding affinities, and the need for protein solubilization [2,3,4]. These challenges can be partly circumvented by expressing soluble ectodomain constructs of membrane proteins [4].

Ectodomains play a crucial role in pathological contexts, as critically underscored by the successful advancement of antibody-mediated blockade therapies [5]. Their exposed location on the cell surface makes them excellent therapeutic targets. However, since the interaction partners of many membrane proteins remain hidden or their interaction potential is not investigated, the full therapeutic potential of ectodomain targeting is likely not yet fully exploited.

High-throughput strategies for uncovering membrane protein interactions usually focus on a single or few proteins or are limited by the sensitivity of methods like mass spectrometry, limiting broader unbiased discovery [6,7,8,9,10]. Systematic screens are rare but powerful [4].

We present a novel recombinant ectodomain protein library with 1784 membrane-associated proteins. Using systematic protein-directed screens, we discovered carbonic anhydrase 9 (CA9) binding to CLEC2 expressed on platelets via an EDxxT motif. This motif has been previously recognized for its role in mediating the binding of the CLEC2 ligands podoplanin (PDPN) and Dectin-1 [11,12]. CA9, upregulated in hypoxic conditions and overexpressed in various cancers, contributes to acidic environments and cancer cell migration and invasion [13,14,15,16,17]. Its limited expression in healthy tissue makes CA9 a promising cancer treatment target [18]. The newfound CA9-CLEC2 interaction highlights CA9′s multifunctional roles and may be crucial for enabling effective CA9-targeted anti-cancer therapies.

## 2. Materials and Methods

### 2.1. Cell Culture of Cell Lines

HCT 116 colon carcinoma cells (ATCC #CCL-247) and HEK-293T cells (ATCC, #CRL-3216) were cultured in McCoy’s 5 A Medium and DMEM (Thermo Fisher Scientific, Karlsruhe, Germany, 16600082 and 61965026), respectively, both supplemented with 10% fetal bovine serum (FBS). HDLM-2 cells (DSMZ, #ACC17) and Ramos-Blue™ cells (Invivogen, Toulouse, France, #rms-sp) were grown in RPMI 1640 Medium GlutaMAX™ (Thermo Fisher Scientific, #61870036) with 20% FBS and 10% FBS plus 0.1 mg/mL zeocin (Gibco Thermo Fisher Scientific, Karlsruhe, Germany, #R25005), respectively. All cells were kept at 37 °C, 5% CO_2_, and split as per the vendor’s instructions and tested negative for mycoplasma contamination.

### 2.2. Plasmid Design

A plasmid library, based on canonical isoform sequences of selected proteins from the UniProt database (June 2021), was created. Sequences were reduced to ectodomains for transmembrane proteins. Retention signals, pro-peptide, and GPI lipidation sites were excluded for ER-Golgi-lysosomal and GPI-anchored proteins, respectively. Constructs were tagged (see results), using either a modified immunoglobulin G1 (IgG1) fragment crystallizable (Fc) region (Fc(dead)) as described previously [19] or a mutated PlGF sequence (mutPlGF) that prevents Flt-1/VEGFR1 [20,21] and NRP-1 [12] binding. Tags contain a short epitope sequence named “2E3” (“ITPPRYRADE”) [22]. The constructs were subcloned into human expression vectors with codon optimization and the addition of a Kozak sequence and restriction (Twist Biosciences and Thermo Fisher Scientific GeneArt).

All other constructs that are not part of the plasmid library were synthesized by Thermo Fisher Scientific GeneArt (Appendix A).

### 2.3. Construct Expression into Conditioned Media of HEK-293T

A recombinant protein library was produced by Trenzyme GmbH (Konstanz, Germany) in 96-well plates using transient transfection of HEK-293T cells with lipofectamine (Thermo Fisher Scientific, #10696153). For hit validation and other binding experiments, transfections were performed in 6-well plates (Thermo Fisher Scientific, #L3000001). Supernatants were collected three days post-transfections.

### 2.4. Protein-to-Cell Interaction Screen

Red blood cells of fresh blood from healthy volunteers, provided by Boehringer-Ingelheim’s company medical officer, were lysed (BioLegend, Koblenz, Germany, #420302), and remaining cells were resuspended in staining buffer (1× PBS 2% FBS, Gibco, #10010023 & #10082147) with Fc receptor blocking reagent (FcR) (Miltenyi Biotec, Bergisch Gladbach, Germany, #130-059-901). 4 × 10^5^ cells per well were seeded into 10 µL of HEK-293T conditioned media with recombinant library proteins in 96 cell-repellent well plates (Greiner Bio-One, Frickenhausen, Germany, #655970) and incubated at 4 °C for 16–24 h. Cells were stained with an antibody panel (Appendix A). This study reports library protein binding to CD14^high^-stained cells. Binding to other cell types diverges from the focus of this study and will be presented separately. After dead cells staining (FVS780, BD, #565388) and washing, samples were resuspended in staining buffer with 1× Qsol solution (Sartorius, Göttingen, Germany, #91161) and analyzed using high-throughput flow cytometry (iQue Screener Plus, Sartorius) and FlowJo™ V10 software. Gating was used to eliminate FVS780 positives, doublets, or aggregates via forward and side scatter of CD14^high^ CD3− stained cells. For the screen, on average, around 3300 events of CD14^high^-stained cell events were recorded per well.

### 2.5. Protein-to-Cell Binding Experiments

Protein-to-cell binding experiments were conducted using flow cytometry analyzed with iQue Screener Plus (Sartorius) or BD LSRFortessa™ cell analyzer. Peripheral blood mononuclear cells (PBMCs) were isolated from blood donation residuals (DRK, Ulm, Germany) using Ficoll-Plaque (Cytiva, Marlborough, MA, USA, #17144003) in SepMate Tubes (StemCell Technologies, Köln, Germany, #85450). Supernatants from SepMate tubes were used as buffy-coat-derived platelet-rich plasma samples (PRP). For platelet preparation, PRP was centrifuged at low (170 g, recover supernatant) and high speed (1000 g, 10 min). PAR1 (protease-activated receptor-1) agonist was used at 8 µM (Abcam Limited, Cambridge, United Kingdom, #ab12801). Monocytes were isolated from PBMCs using the EasySep Human Monocyte Cell Isolation Kit (StemCell Technologies, Köln, Germany, #19359RF), with the platelet removal cocktail added or omitted as required. Red blood cells were lysed with RBC lysis buffer (eBioscience Thermo Fisher Scientific, Karlsruhe, Germany, #00-4333-57). Primary Ramos-Blue™ cells or HCT 116 cells were resuspended in staining buffer for binding tests. Tests with whole blood samples of mice were performed in the presence of 2.5 mM Gly-Pro-Arg-Pro (Sigma Aldrich, Schnelldorf, Germany #G5779). HCT 116 cells were treated for 18 h at 37 °C with neuraminidase from *Vibrio cholerae* (Roche Diagnostics GmbH, Mannheim, Germany, #11080725001) prior to detachment (TrypLE™, Gibco Thermo Fisher Scientific, Karlsruhe, Germany, #12604013) and incubation with HEK-293T-conditioned media. At least 5000 events were recorded for each flow cytometry experiment. Forward and side scatter were used to eliminate doublets or aggregates. FVS780-stained cells were excluded for analysis. Proteins and staining reagents are listed in the Appendix A.

### 2.6. AlphaLISA

CA9^38−414^-His-avi (60 nM, Acro Biosystems, Basel, Switzerland, #CA9-H82E3) coated streptavidin AlphaSCREEN donor beads (20 ug/mL, Revvity, Hamburg, Germany, #6760002) were added to 2 µL of conditioned media recombinant library ectodomains. Binding is detected by the addition of AlphaLISA protein A acceptor beads (10 ug/mL, Revvity, #AL101R) coated with an anti-2E3 antibody22 (0.5 nM). Screening assays were performed in 1536-well plates (Greiner Bio-One, Frickenhausen, Germany, #782075), with a final volume of 3 µL in AlphaLISA Immunoassay Buffer (Revvity, #AL000F). Confirmation experiments were performed in 384-well plates (Greiner Bio-One, Frickenhausen, Germany, #784075) with a final volume of 10 µL using 7 µL of conditioned media ectodomains. Acceptor bead activation indicates proximity to the donor bead and thus, potential binding events. Signal was detected using a PheraSTAR FSX plate reader (BMG Labtech, Freiburg, Germany) using 680 nm excitation and 615 nm emission wavelength.

### 2.7. Simple Western Immunoblots

Simple Western immunoblots were conducted using Peggy Sue (bio-techne, ProteinSimple, Minneapolis, Minnesota, USA) and the Size Separation Master Kit with Split Buffer (12–230 kDa) as per the manufacturer’s guidelines. Supernatants from transiently transfected HEK-293T cells were diluted 1:20 for simple Western. For anti-Syk/Phospho-Syk (cell signaling, Leiden, Netherlands, #2712S and #2711S), simple Western, Ramos-Blue™ and HCT 116 cells (9:1 ratio) were incubated for 15 min at 37 °C. The reaction was stopped with ice-cold RIPA buffer (Cell Signaling, #9806) with protease/phosphatase inhibitor cocktail (Thermo Fisher Scientific, #1861281). Protein expression was assessed through images of individual lanes, and Phospho-Syk peak area was quantified using Compass software (ProteinSimple, Compass for SW Version 5).

### 2.8. Cell Line Generation

Ramos-Blue™ cells were transfected using the SG Cell Line 4D-Nucleofector™ kit (Lonza Bioscience, Basel, Switzerland, #V4XC-3024) and HCT 116 with Lipofectamine 3000 in OptiMEM (Gibco #11520386, Thermo Fisher Scientific, #L3000001). Cells were selected with Blasticidin S HCl (Ramos: 25 µg/mL; HCT 116: 10 µg/mL (Gibco™, #A1113903)) and sorted for high expression using the Sony MA900 sorter.

### 2.9. Analysis of Cell Surface Protein Expression in Cell Lines

Ramos-Blue™ and HDLM-2 cells were collected, and HCT 116 cells were detached (TrypLE™, Gibco™ #12604013) and collected. Cells were resuspended and stained (Appendix A) in staining buffer with FcR. Post-washing, cells were analyzed using a BD LSRFortessa™ cell analyzer and FlowJo™ V10 software, with gating based on cell size (FSC and SSC) to exclude doublets, aggregates, electronic noise, and dead cells (FVS780 stained).

### 2.10. Reporter Cell Assays

Ramos-Blue™ cells were stimulated either with soluble protein or through co-culture with confluent HCT 116 or HDLM-2 cells. Ramos-Blue™ wildtype (WT) or CLEC2^2E3^-expressing cells (3.42 × 10^5^/well) were added without FBS/antibiotics (0.5 µM BI-1002494 or DMSO added as indicated). Post-overnight culture, phosphatase activity was measured according to the manufacturer’s instructions (Invivogen, #rep-qbs) with a Tecan microplate reader. Each experiment was performed in triplicate and repeated independently at least three times per condition.

### 2.11. Statistical Analysis

Data plotting and statistical analysis were performed using GraphPad Prism V10. Details about the statistical tests, error bars, and biological replicates can be found in the figure legends. Flow cytometric scatter plots and histograms were created using FlowJo™ V10.

## 3. Results

### 3.1. Recombinant Membrane Protein Library Screen Identified Interaction Between CA9 and CD14^high^ Stained Blood Cells

We created a recombinant protein library with 1504 single-pass membrane proteins, 115 multi-pass membrane proteins, 131 GPI-anchored proteins, and 34 ER retention signal proteins (Figure 1A). The proteins were engineered to express either their linear ectodomains or as full-length proteins without GPI-anchor sites or ER retention signals. For multi-pass proteins, only the extracellular sequence before the first transmembrane domain was included if it was longer than 20 amino acids (aa) (including the signal sequence if present).

Proteins with N-terminal extracellular domains were C-terminally tagged with a modified immunoglobulin G1 (IgG1) fragment crystallizable (Fc) region (Fc(dead)) as described previously [19]. Alternatively, proteins with C-terminal extracellular domains were N-terminally tagged with a mutated PlGF sequence (mutPlGF) that prevents Flt-1/VEGFR1 [20,21] and NRP-1 [12] binding. Both tags are extended with a short epitope sequence named “2E3” (“ITPPRYRADE”) [22]. GPI-anchored and ER retention signal proteins were also C-terminally tagged with Fc(dead)^2E3^. All library proteins homodimerize via Fc(dead) or mutPlGF and can be detected when bound to surfaces via their 2E3 peptide tags [22] (Figure 1A and Figure 3A).

Individual library proteins in HEK-293T conditioned media were employed for a protein-to-cell flow cytometry screen (Figure 1B,C), which identified binding of several ectodomain library proteins to CD14^high^-stained cells (Figure 1D and Appendix A). The mean fluorescence intensity in the screen depends on numerous factors, including affinity, receptor density, and protein expression levels of each library construct. We therefore arbitrarily selected a cut-off that included numerous hits well known for their ability to bind to monocytes (e.g., CD47, CD6, X3CL1), resulting in an MFI signal of 1.5× greater than the average MFI of negative controls. None of the negative controls exceeded 1.5 MFI (Appendix A).

Unexpectedly, we identified an interaction of carbonic anhydrase 9 (CAH9/CA9) with CD14^high^ cells.

CA9’s known primary role is to regulate cellular pH balance and to be critical for the growth of cancer cells in acidic or low-oxygen environments [23,24], which must cope with hypoxia and acidosis. CA9 is currently under investigation as a potential therapeutic target, with ongoing preclinical and clinical trials. Any new insights into its functions could be relevant for CA9-directed therapeutics. Considering that CA9 is strongly upregulated in many cancers and no direct interaction of CA9 with peripheral blood cells has been reported so far, we decided to further explore this interaction.

### 3.2. CA9 Binding Is Detected to Platelets but Not Isolated Monocytes

In our screening setup, we detected binding of numerous proteins known to interact with monocytes, such as TNFA, CXCL3, SHPS1, TYRO3, LYAM3, and CD2 (Figure 1A). Exemplary testing confirmed COL12 and CD47 binding to purified monocytes, as expected (Figure 2A). We were also able to confirm the binding of CA9 to CD14^high^ cells but not to CD3+, CD19/20+, CD66c+, or CD56+ cells using whole blood samples (Appendix A). However, this interaction was weak if tested with isolated PBMCs (Appendix A) and essentially lost if tested with purified monocytes (Figure 2A). Consistently, CA9 binding to a monocyte-like cancer cell line (THP-1) was not detected (Appendix A). Incomplete removal of platelets during monocyte preparation partially rescued binding, raising the question if indirect interaction of CA9 with monocytes via platelets could explain CA9 binding in whole blood (Appendix A). Indeed, we detected CA9 binding only to CD14^high^ CD41a+ but not CD41a− cells (Figure 2B), and the CA9 library construct bound to CD42a+ buffy coat (BC)-derived platelets (Figure 2C).

CA9 was not the only protein that bound indirectly to monocytes via platelets. For instance, we could not confirm JAM2 and OLR1 binding to isolated monocytes but detected binding to platelets (Figure 2A and Appendix A).

CA9 binding to platelets was confirmed using commercially available purified CA9^38−414^-His-avi (Figure 2D). CA9 binds to CD62P− and CD62P+ platelets with better binding capacity to CD62P+ platelets (Appendix A). Unlike PAR1 agonist treatment, the ratio of CD62P+/CD62P− platelets remained unchanged between CA9-treated and untreated BC-derived platelets, indicating that soluble CA9 (sCA9) is unable to activate platelets (Appendix A). Additionally, murine sCA9^his^ binds to mouse platelets, proving the evolutionary conservation of the interaction (Appendix A).

### 3.3. Identification of a Mammalian CA9 Platelet Binding Region

To identify the platelet binding site within CA9, we designed C-terminally tagged truncated CA9-mutPlGF constructs that included an extended part of the catalytic domain (aa. 113–391; CA^tag^) or the proteoglycan (PG)-like domain (aa. 38–112, PG-short^tag^) (Figure 2E). Despite successful expression, we were unable to detect binding of the CA^tag^ and PG-short^tag^ constructs to platelets (Figure 2F,G). A redesigned, slightly longer version of the PG-short construct (PG-long^tag^, aa. 38–137) was also tested (Figure 2E). With PG-long^tag^, CA9 binding to platelets was recovered, suggesting that the platelet binding region may include a small part of the Uniprot-annotated PG-like domain and the linker region between the PG-like and CA domains (Figure 2E–G).

The putative platelet binding region sequence was similar to CA9 from numerous mammals, and interestingly, for species like chickens, which lack platelets, we could not identify a similar sequence homology (Figure 2E). Based on homology, we designed a new 41 aa. Peptide, CA9-96-139^tag^, which showed similar binding to platelets (Figure 2E–G). Conversely, replacing a 14 aa. core (aa. 105–119) with a glycine-serine peptide in the full-length CA9 ectodomain was sufficient to ablate the platelet interaction (Appendix A).

In conclusion, we here identified a platelet-binding region within CA9 that is conserved in mammals and includes the C-terminal end of the PG-like domain and the intervening sequence linking the PG-like and CA domains.

### 3.4. An AlphaLISA Screen Identifies Novel CA9 Binding Partners

We established an AlphaLISA screen using conditioned media, leveraging our library of recombinant proteins, to identify potential CA9^38−414^-His-avi binding partners on the platelet surface (Figure 3A).

The AlphaLISA setup was validated using tagged versions of Girentuximab (G250) vs. a randomly selected control protein from our library (CONA1) (Appendix A). We detected binding of both G250 constructs to CA9^38−414^-His-avi, but not of the control ectodomain (Appendix A). Although lower concentrations of CA9 on the beads (15 nM) were sufficient for binding detection, more robust results were obtained with higher concentrations (60 nM) and selected for the screen.

The AlphaLISA screen detected a >650-fold signal increase above background signal for GLT16 (GALNT16), CLC1B (CLEC2), and LYAM3 (P-selectin) (Figure 3B and Appendix A). We focused on CLEC2 and LYAM3 since they are well-known platelet surface proteins and validated the screening result (Figure 3C,D).

### 3.5. CA9 and Podoplanin Share Sequence Homology

We performed sequence comparisons of CA9 and Podoplanin (PDPN), a known endogenous ligand of CLEC2 [25], and found they share a short but identical “EDLPT” sequence (CA9-111-115/PDPN-81-85). Notably, CA9-Thr115, like these PDPN sequences, is also reportedly glycosylated [26]. Nearby regions of both these proteins possess repetitive sequence elements and are enriched in acidic aa. (Figure 4A), which may interact with the arginine patch surrounding the ligand binding site of CLEC2 [27,28]. PDPN has two known O-linked glycosylation sequences, Thr52 “EDDVVT” and Thr85 “EDLPT”, but Thr85 is described to be the major site for CLEC2 binding [25,29,30,31]. Moreover, an EDxxT motif was also found in human Dectin-1 and has been reported to mediate Dectin-1-CLEC2 interaction [12]. The presence of an EDxxT motif in CA9 and the similarities with PDPN encouraged us to further investigate the putative CA9/CLEC2 interaction. Unlike PDPN, CA9 likely possesses only a single CLEC2 binding site as indicated by the absence of a second glycosylated threonine. The presence of soluble PDPN^ectodomain^ (sPDPN^23−131^-His) impairs platelet binding of biotinylated sCA9^38−414^-His-avi in a dose-dependent manner, suggesting competition for the same binding receptor site on platelets (Figure 4B).

### 3.6. Thr115 Is Required for CA9 Binding to Platelets

We designed tagged CA9^96−137^-T115A and CA9^96−137^-TVE-to-NVS-115-117 (encoding N-linked instead of O-linked glycosylation [32]) mutant constructs to test if O-linked glycosylation of Thr115 is important for platelet binding [26]. Binding to platelets was not detected for both mutant constructs despite similar expression levels as the control construct (Figure 4C,D). The CLEC2 construct from the recombinant library was also found to bind to HCT 116 cells overexpressing a non-shed version of CA9^NS^ or PDPN [33] (Figure 4E and Figure 5D). Overnight treatment of these HCT 116 cells with neuraminidase ablated CLEC2 binding, indicating that sialylation of Thr115 is required for CLEC2 binding. This provides further evidence that the sialylation of the EDxxT motif is necessary for CLEC2 interaction [12,30,31,34].

### 3.7. CA9 Signaling via CLEC2 Is Syk-Dependent, Requires Thr115 and Is Regulated by Shedding

CA9-CLEC2 binding was validated using a CLEC2^2E3^ overexpressing Ramos-Blue™ cell line. We detected binding of sCA9^38−414^-His-avi to this cell line but not to WT Ramos-Blue™ control cells lacking any detectable CLEC2 expression (Figure 5A,B). CA9 binding to CLEC2^2E3^ overexpressing cells was competed in the presence of sPDPN^23−131^-His or soluble His-CLEC2^55−229^ (Figure 5B).

We next asked if CA9 could induce CLEC2 intracellular signaling as reported previously for PDPN [34]. RNA-sequencing data suggest that proteins relevant for CLEC2 signaling are expressed in Ramos cells (Figure 5A) [35,36]. Using engineered CLEC2^2E3^-expressing Ramos-Blue™ reporter cells, we detected a small increase in NF-κb activation when incubated with high concentrations of sCA9^38−414^-His-avi or sPDPN^23−131^-His (Figure 5C).

CLEC2 activation requires effective receptor clustering [37]. To improve CLEC2 clustering, we generated stable HCT 116 cell lines with ^2E3^CA9 or ^2E3^PDPN overexpression and co-cultured HCT 116 cells overnight with CLEC2^2E3^ or WT reporter cells. Both ^2E3^CA9 and ^2E3^PDPN but not WT HCT 116 cells activated Ramos-Blue™ CLEC2^2E3^ but not WT cells. Activation by HCT 116 ^2E3^PDPN or ^2E3^CA9 was completely inhibited in the presence of 0.5 µM BI-1002494 spleen tyrosine kinase (Syk) inhibitor [38], indicating that reporter cell activation was mediated via the previously described Syk-dependent pathway [39] (Figure 5D,E).

The PDPN-expressing HCT 116 cells activated Ramos-Blue™ CLEC2^2E3^ more effectively compared to CA9 overexpressing HCT 116 cells (Figure 5E). The weaker CA9-mediated signaling may be the result of the absence of a second CLEC2 binding site on CA9. In addition, CA9 shedding by ADAM10/17 [40,41] could negatively impact CLEC2 clustering. To prevent shedding, we generated an HCT 116 cell line with overexpression of a CA9-non shed (CA9^NS^) version [33] (Figure 5D), which statistically significantly improved reporter cell activation (Figure 5E). We confirmed CLEC2 stimulation with HCT 116 CA9^NS^ cells via Syk by anti-Phospho-Syk immunoassay in Ramos-Blue™ CLEC2^2E3^ but not WT cells (Figure 5F,G).

We also generated an HCT 116 cell line with overexpression of a CA9-T115A full-length construct. Despite similar expression levels to HCT 116 CA9^wt^ or CA9^NS^ cells, reporter cell activation is not observed when stimulated with HCT 116 CA9-T115A cells (Figure 5D,E).

In conclusion, CA9 instigates CLEC2 signaling by binding to CLEC2 via its glycosylated Thr115 residue. CLEC2 activation is attenuated by CA9 shedding.

### 3.8. A Lymphoma Cell Line Can Activate CLEC2

Loss-of-function mutations in the Von Hippel-Lindau gene are frequently found in clear cell renal cell carcinoma (ccRCC) leading to CA9 overexpression [42]. Since kidney cells, however, potentially express PDPN, we sought to identify a cancer cell line with CA9 but not PDPN under normoxic conditions as found in peripheral blood where platelets are predominantly found. While CA9’s role in regulating cellular pH balance in solid tumors is well described, we hypothesize a different function for hematological cancer, potentially related to CLEC2 interaction. Expression of CA9, but not PDPN, is found for several lymphoma cell lines (Figure 6A) [35]. We confirmed this for the Hodgkin lymphoma-derived cell line HDLM-2 at the protein level (Figure 6A). HDLM-2 cells can induce NF-κb activation in our reporter cells with CLEC2 overexpression but not in WT reporter cells (Figure 6B).

### 3.9. Genetic Lead Variants Linked to GALNT16 Gene Loci Associate with Platelet Traits

Lastly, we investigated variant-gene-trait associations for GALNT16 (GLT16). This protein demonstrated the most potent interaction signal in the AlphaLISA screen and was the sole protein among the 18 UDP-N-α-D-galactosamine polypeptide N-acetylgalactosaminyltransferases (GALNT) to bind with CA9 (Figure 3B). A comparative analysis of various genome-wide association studies (GWAS) led us to identify 29 different lead variants for GALNT16 associated with platelet-related traits. Notably, compared to other traits linked with GALNT16 variants, the associations with platelet-related traits were the most significant, as indicated by the lowest p-values. Eleven of these lead variants were also replicated in multiple studies (Appendix A). In summary, GWAS data strongly suggest a platelet-related role for GLT16 in humans. Its ability to bind to CA9 implies a potential link between GLT16-mediated modification of CA9 and CA9-CLEC2 interaction.

## 4. Discussion

We have designed a comprehensive recombinant protein library and pioneered a strategy of two sequential interaction screens. First, we discovered CA9-platelet interaction using a systematic protein-to-cell binding screen (Figure 1). Subsequently, a CA9-directed interaction screen identified its binding to CLEC2 (Figure 3).

Our first screen detected several expected interaction events, including the binding of LYAM3, CD2, TYRO3, SHPS1, or TNFA to monocytes [43,44,45,46,47]. For three exemplary known binders (CD47, COL12, and GSGL1), we confirmed interaction with either purified monocytes or THP-1 cells (Figure 2A and Appendix A). With this screening approach, protein binding was also detected to monocyte-associated platelets, albeit at the expense of needing to deconvolute the target cell in confirmation experiments as demonstrated for CA9, JAM2, and ORL1 (Figure 2A and Appendix A).

The inherent protein expression by HEK-293T cells could negatively impact the detection of protein-to-cell interactions and molecular interactions detected in our AlphaLISA assay. Moreover, our screens cannot rule out indirect binding of protein constructs via endogenously expressed HEK293-T or FBS proteins. For example, Collectin12 (COL12), found to bind to monocytes (Figure 1D and Figure 2A), could theoretically bind indirectly to monocytes via C-reactive protein [48]. Furthermore, the binders identified in this screen are likely biased towards the dominant CD14^high^ population. This could potentially skew the results and limit the diversity of the binders identified. Protein binding to subpopulations that are relatively underrepresented in the peripheral blood was likely missed in this screen. This could further lead to an incomplete understanding of the full range of protein-cell interactions occurring in peripheral blood. Future recombinant protein-to-cell with isolated cell populations may be required to identify overlooked interactions within subpopulations.

We paid particular attention to the discovery of CA9 binding to platelets due to its significant role for many cancers and therapy, as well as the lack of a known immune cell CA9 receptor [15,18]. While CA9^ectodomain^ is proposed to form a complex with ion transporters likely via its catalytic domain [49,50,51], our findings showed that the catalytic domain was dispensable for platelet binding (Figure 2E–G). Platelet binding was also not detected for CA9’s unstructured PG-like domain, proposed to be a protein binding hub [52], but no binding partners have been unequivocally assigned. In contrast, we identified the previously unannotated sequence linking the PG-like and catalytic domains as the crucial interaction site for CLEC2 (Figure 2E–G). Specifically, Thr115, known to be glycosylated [26], is required for CA9-platelet interaction and receptor activation (Figure 4C,D and Figure 5E).

The CA9-directed AlphaLISA screen identified potential CA9 interaction partners, notably GLT16, CLEC2, and LYAM3. While we focused on CA9-CLEC2, separate investigations for CA9-LYAM3 and CA9-GLT16 should be pursued in the future. Notably, GALNT16 gene variants associate with platelet traits, supporting the notion that GLT16 may mediate O-linked glycosylation of CA9-Thr115 (Appendix A).

Increased binding signals in our screen were found for two complement receptors (CR). However, binding of CA9 to CR1/2-expressing immune cells was not confirmed (Figure 2A and Appendix A). These interactions could be mediated by bridging molecules or may be restricted to soluble CR receptors. Given the CA9-platelet binding and previously proposed crosstalk between the coagulation and complement systems [53], a potential CA9 link to the complement system warrants future investigation.

PDPN and CA9 show striking sequence similarities and functional overlap in vitro (Figure 4A and Figure 5E–G). PDPN-overexpressing HCT 116 cells were more potent CLEC2 activators than those overexpressing CA9^wt^. This difference may be explained by either the lack of a second CLEC2 interaction domain in CA9, slightly differing expression levels, or higher CA9 shedding from the cell surface [33,40,41]. HCT 116 cells expressing CA9^NS^ activated CLEC2 signaling more effectively than CA9^wt^ cells, supporting the importance of shedding to regulate CLEC2 activation (Figure 5E). This may be unique to CA9 since PDPN shedding is not reported to our knowledge.

The PDPN–CLEC2 interaction can enhance tumor growth and metastasis and could cause thromboembolism in cancer patients by promoting platelet-tumor cell aggregation and releasing pro-tumorigenic factors [11,36,54,55,56,57,58,59]. Given the similarities between CA9 and PDPN, we propose a similar role for the CA9-CLEC2 interaction. We demonstrated that Hodgkin lymphoma cells, which endogenously express CA9, but not PDPN, induce CLEC2 signaling (Figure 6A,B). Altered platelet biology, a negative prognostic factor in advanced Hodgkin lymphoma [60], may be influenced by CA9-mediated activation of platelets or megakaryocytes. Additionally, while most models describe CA9’s role in solid tumors, its function in hematological malignancies has been suggested but remains poorly understood [61]. The discovery of CLEC2 interaction describes a new putative mechanism for CA9’s function in hematological cancers or circulating cancer cells (CTC). Moreover, potential PDPN and CA9 redundancy may limit the response to anti-CA9 therapy in ccRCC, where both proteins are expressed [18]. The treatment can also cause severe thrombocytopenia [18], possibly related to CA9–platelet interaction.

CA9 is highly stable and can be released into the extracellular space via ectodomain cleavage or exosomes [17,33,34,35,36,37,38,39,40,62]. Elevated sCA9 plasma levels (pg to ng/mL [63,64]), triggered by ADAM17 upregulation under hypoxia [65], raised the possibility of cell contact-independent activation of CLEC2. However, our investigations showed that even high concentrations of sCA9 or sPDPN (20 µg/mL) showed only weak CLEC2 activation, indicating that these soluble ligands are at best weak activators (Figure 5C). Enhanced CLEC2 activation by HCT 116 cells expressing CA9^NS^ instead of CA9^wt^ (Figure 5C–E) lends further support to these observations. Consistently, CLEC2 activation in human platelets was reported with HEK-293T cells, but not with soluble Fc-PDPN [37]. In mouse models, CA9^NS^ expression in cancer cells increased metastases, strongly supporting a functional difference between soluble and membrane-bound CA9 [33]. The enhanced ability of CA9^NS^ to activate CLEC2 may promote platelet-mediated CTC protection in addition to previously described improved migratory properties provided by CA9 [13,14,66]. Interestingly, the interaction of an unknown ligand with CLEC2 is reported to contribute to thrombus stability, independent of CLEC2 downstream signaling [67,68]. If sCA9 can fulfill this role, it deserves future investigation.

CA9 expression is largely restricted to a few tissues, and gastric hyperplasia is reported in CA9 knock-out mice [69,70]. A blood–lymph mixing phenotype, similar to that seen in PDPN knock-out, is not reported for CA9 [71]. However, the release of platelet-derived factors, potentially induced by CLEC2 activation via CA9, may contribute to normal gut development and warrants further investigation.

However, in pathological conditions, the newly discovered CA9-CLEC2 interaction may also play a role in diseases other than cancer. For instance, rheumatoid arthritis (RA) synovial fibroblasts upregulate CA9 up to 28-fold under hypoxia, a hallmark of RA [72]. Along with PDPN, which is also reported to be highly upregulated in RA, CA9 may contribute to the induction of the thrombo-inflammatory role of platelets and deserves further investigations [73,74].

In conclusion, we provide a comprehensive dataset of proteins binding to monocytes or platelets. We identified the binding of CA9, a gene strongly expressed in response to hypoxia in cancer, to platelets. We validated selected CA9 interaction partners and showed that membrane-bound CA9 binds and activates the platelet immunoreceptor CLEC2. Striking structural and functional similarities with PDPN indicate that the newfound CA9-CLEC2 interaction may be important for future anti-cancer therapies and could contribute to tumor growth and metastasis.

## Figures and Tables

**Figure 1 cells-13-02083-f001:**
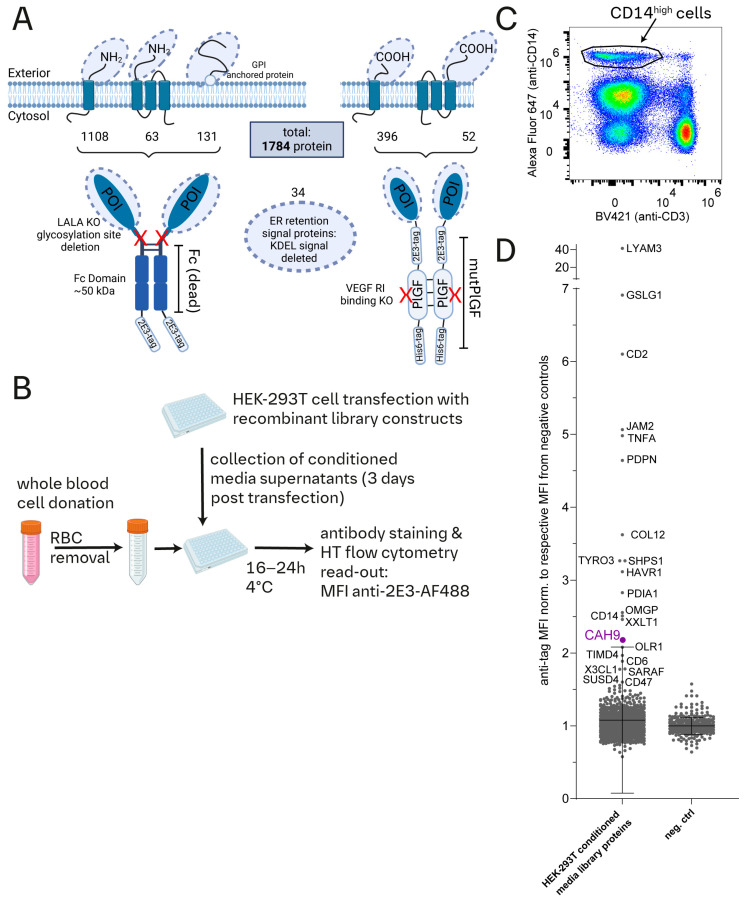
A conditioned media recombinant protein library to identify protein binding to CD14^high^-stained cells of peripheral blood. (**A**) A schematic illustration outlining the design of the recombinant protein library (created with BioRender). Protein selection is based on uniprot annotations and includes the ectodomains of type I and II, as well as selected multi-pass transmembrane (TM) proteins. The library also comprises ectodomains of glycosylphosphatidylinositol (GPI)-anchored proteins and proteins without annotated TM domains but with an endoplasmic reticulum (ER) retention signal sequence. Depending on the protein architecture, truncated proteins (protein of interest, POI) are either C-terminally tagged with a mutated human Fc domain (Fc(dead)) followed by a short tag peptide sequence named “2E3” (“ITPPRYRADE”), or N-terminally tagged with a mutated human PlGF combined with the 2E3 and a 6xHis tag (mutPlGF). These protein tags enable the dimerization of library proteins. The numbers of included proteins are noted beneath the schematic representation of each protein type. (**B**) A schematic representation of the protein-to-cell binding screen. Following red blood cell lysis of peripheral blood from healthy human donors, cells are added to conditioned media recombinant library harvested from transfected HEK-293T cells in a 96 well plate format. After 16–24 h incubation, cells are stained and analyzed by high-throughput flow cytometry. Protein-to-cell binding is assessed by quantification of the median fluorescence intensity (MFI) of Alexa Fluor 488 conjugated anti-2E3 obtained by flow cytometry. (**C**) Example of flow cytometric analysis of peripheral blood cells stained to identify CD3− CD14^high^-expressing cells. (**D**) Protein-to-CD14^high^ screening result. Alexa Fluor 488 MFIs of CD14^high^ stained cells were normalized to the mean MFI obtained with negative controls. The graph shows normalized MFI for conditioned media library protein or negative controls. The MFI of carbonic anhydrase 9 (CAH9/CA9) sample is highlighted in purple and exhibits a 2.2-fold increase relative to the mean MFI of negative controls.

**Figure 2 cells-13-02083-f002:**
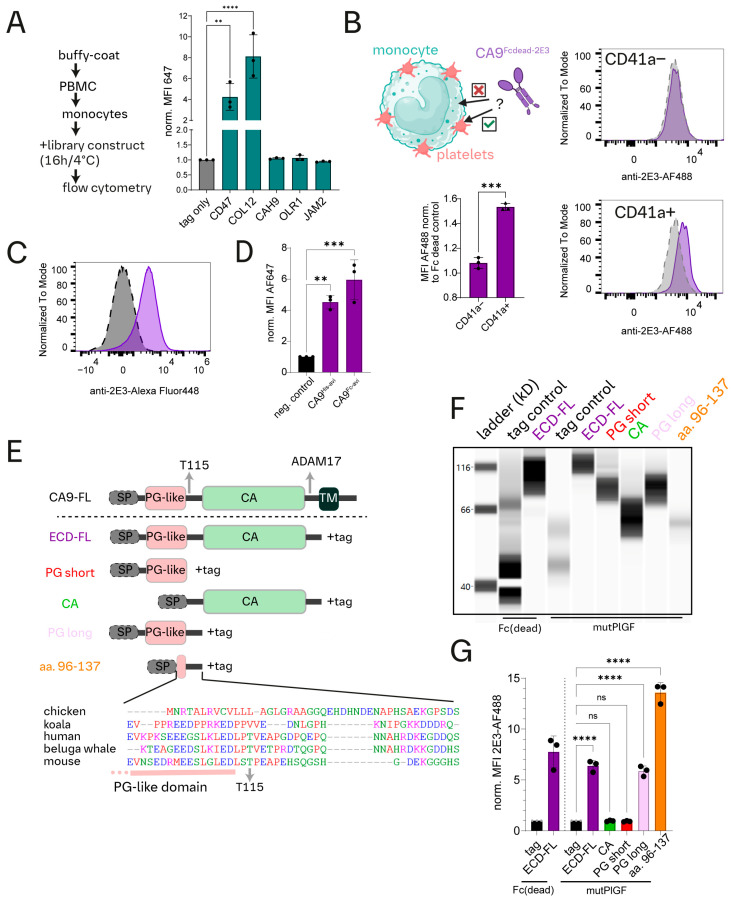
CA9^ectodomain^ binds indirectly to monocytes via monocyte-associated platelets. (**A**) Binding tests of selected constructs from the recombinant protein library, expressed in conditioned media (HEK-293T cell expression), to isolated monocytes. Unlike the whole blood mixed cell population, an MFI increase compared to a tag control (^6xHis−2E3^Fc(dead)) was not detected for library constructs CAH9, OLR1, and JAM2. Library constructs for CD47 and COL12 were used as positive controls for binding to isolated monocytes. Each dot represents one donor. Statistical analysis: one-way ANOVA, Dunnett’s multiple comparison test. ** = 0.0073, **** < 0.0001. (**B**) Top left: A schematic representation of the hypothesis for indirect binding of CA9 to monocytes (CD14^high^) via platelets as indicated by a red “×” for no binding to monocytes and a green check mark for binding to platelets. Right: An example of the flow cytometry results for CA9 binding to CD14^high^ CD41a− or CD14^high^ CD41a+ cells. Whole blood samples were incubated either with conditioned media containing human ^6xHis−2E3^Fc(dead) control construct (grey dashed line) or CA9-Fc(dead)^2E3^ construct (purple solid line). Lower left panel: Quantification of MFI fold change for binding of CA9-Fc(dead)^2E3^ compared to tag control for platelet positive and negative monocytes. *** = 0.001, unpaired *t* test. (**C**) Flow cytometry result of ^6xHis−2E3^Fc(dead) (black dashed line) vs. CA9-Fc(dead)^2E3^ (purple solid line) binding to CD41a stained buffy-coat derived platelets. (**D**) Binding test of purified CA9^38−414^-His-avi or CA9^38−414^-Fc-avi to buffy-coat-derived platelets (16 h incubation at 37 °C). Quantification of MFI fold change compared to no protein negative control. Statistics: ** = 0.0027, *** = 0.0004; one-way ANOVA, Dunnett’s multiple comparison test. Platelets from 3 healthy donors are compared. (**E**) A schematic representation of CA9 domain structure (SP = signal peptide, PG = proteoglycan, CA = catalytic domain, TM = transmembrane domain). Truncated CA9 constructs used in experiments shown in (**F**,**G**). Partial sequence alignment of CA9 from different organism shows conservation of several amino acids within the platelet binding region of CA9 in mammals (Mus musculus (mouse), Phascolarctos cinereus (koala), Delphinapterus leucas (Beluga whale)) but not in chickens (Gallus gallus). (**F**) Simple protein immunoassay results to assess the expression of CA9 or tag constructs. Detection with anti-2E3 antibody. (**G**) Conditioned media binding test of CA9 constructs to CalceinRed-Orange stained buffy-coat derived platelets. Quantification of MFI fold change compared to tag control constructs. Statistics: ns = not significant, **** < 0.0001; one-way ANOVA, Dunnett’s multiple comparison test. Each dot represents one donor.

**Figure 3 cells-13-02083-f003:**
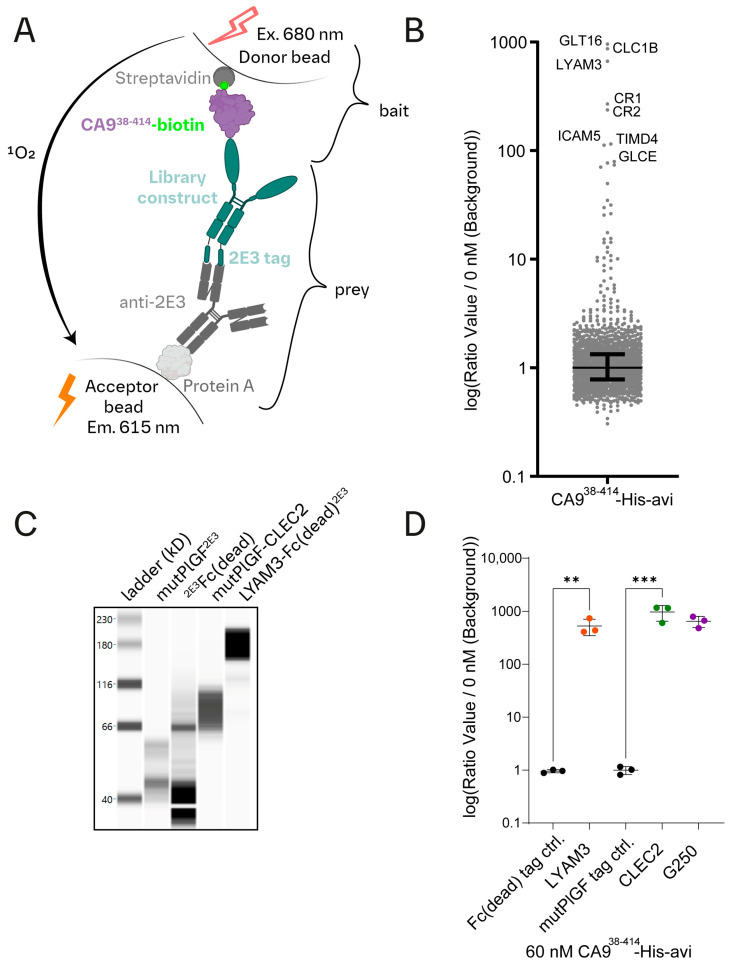
An AlphaLISA CA9^ectodomain^ interaction screen identifies novel interaction partners. (**A**) Schematic representation of the AlphaLISA setup to identify CA9 binding proteins within the recombinant protein library (created with BioRender and Adobe Illustrator). (**B**) AlphaLISA conditioned media screening results. Given the large number of tested constructs, the median emission signal of all constructs of an individual 1536 assay plate was employed as the background signal. For each ectodomain, the emission signal was measured in duplicates, and the median signal is normalized to the background and represented as a dot in the graph. Black lines represent the median and interquartile range. (**C**) Simple protein immunoassay to assess the expression and secretion of indicated protein constructs by HEK-293T cells into the supernatant (3 days post-transfection). (**D**) AlphaLISA CA9^38−414^-His-avi binding validation experiment. A CA9-directed AlphaLISA experimental setup was applied to test for CA9^38−414^-His-avi binding to recombinant library ectodomains (LYAM3 and CLEC2). Respective tag controls for type I (^6xHis−2E3^Fc(dead)) or type II (^6xHis^mutPlGF^2E3^) membrane proteins were used as negative controls. Anti-CA9 (2E3 tagged Girentuximab (G250)) was used as a positive binding control. Emission signals were normalized to control condition containing HEK-293T-conditioned media of non-transfected HEK-293T cells. Statistics: ** = 0.0098 and *** = 0.0001; one-way ANOVA, Šídák’s multiple comparisons test. Each dot represents an independent experiment with triplicate measurement.

**Figure 4 cells-13-02083-f004:**
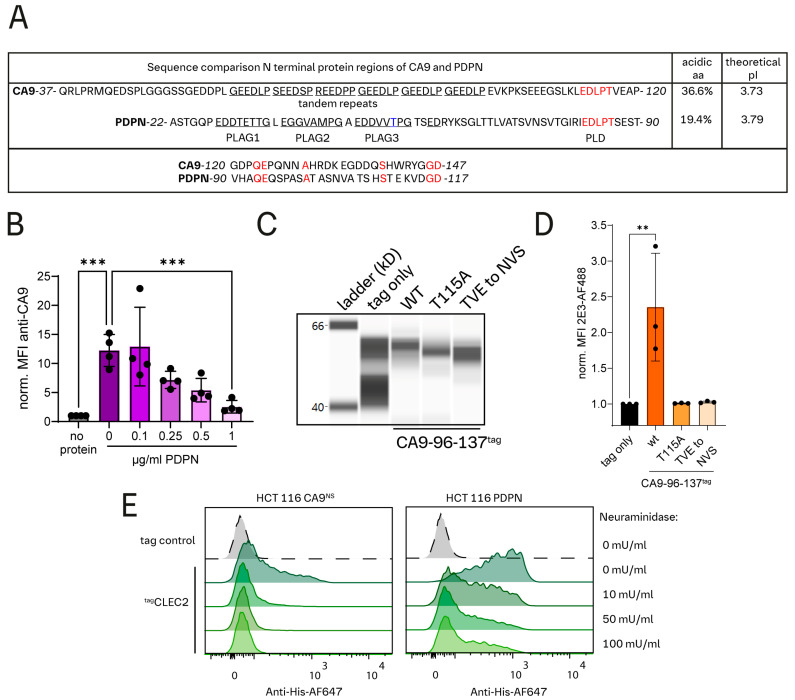
Sequence similarities and platelet binding competition between CA9 and PDPN. (**A**) Comparison of the N-terminal sequences of CA9 and PDPN. Both proteins share an “EDLPT” motif (highlighted in red) within this sequence section, possess repetitive elements (underlined in the figure) upstream of the “EDLPT” sequence, and have an accumulation of acidic amino (aa) acids (low theoretical isoelectric point (pI)). Sequence homology is also found downstream of the EDLPT motif. While PDPN has been shown to have two CLEC2 interaction domains (platelet aggregation stimulating domain (PLAG3) with a glycosylated threonine highlighted in blue and PLAG-like domain (PLD), CA9 likely only has one CLEC2 binding domain since a second similar sequence providing the ability for O-linked glycosylation is absent. (**B**) Binding competition test of CA9 and PDPN. Binding of biotinylated CA9^38−414^-His-avi (1.5 µg/mL) was tested to buffy-coat-derived platelets in presence or absence of PDPN^23−131^-His. Quantification of MFI fold change compared to a negative control (no protein) is shown. *** = 0.002 and *** = 0.008; one-way ANOVA Šídák’s multiple comparisons test. Each dot represents one donor. (**C**) Simple protein immunoassay to assess expression of indicated protein constructs by HEK-293T cells into the supernatant (3 days post-transfection). (**D**) Conditioned media binding test of CA9 constructs to buffy-coat derived platelets. The quantification of MFI fold change compared to tag control constructs is shown. ** = 0.0058; one-way ANOVA, Dunnett’s multiple comparison test. Each dot represents one donor. (**E**) Binding test of mutPlGF-CLEC2, expressed in HEK-293T conditioned media, to HCT 116 cells with either CA9^NS^ or ^2E3^PDPN expression, as determined by flow cytometry. HCT 116 cells were cultured overnight, with or without neuraminidase, prior to incubation with the conditioned media at 4 °C, followed by staining.

**Figure 5 cells-13-02083-f005:**
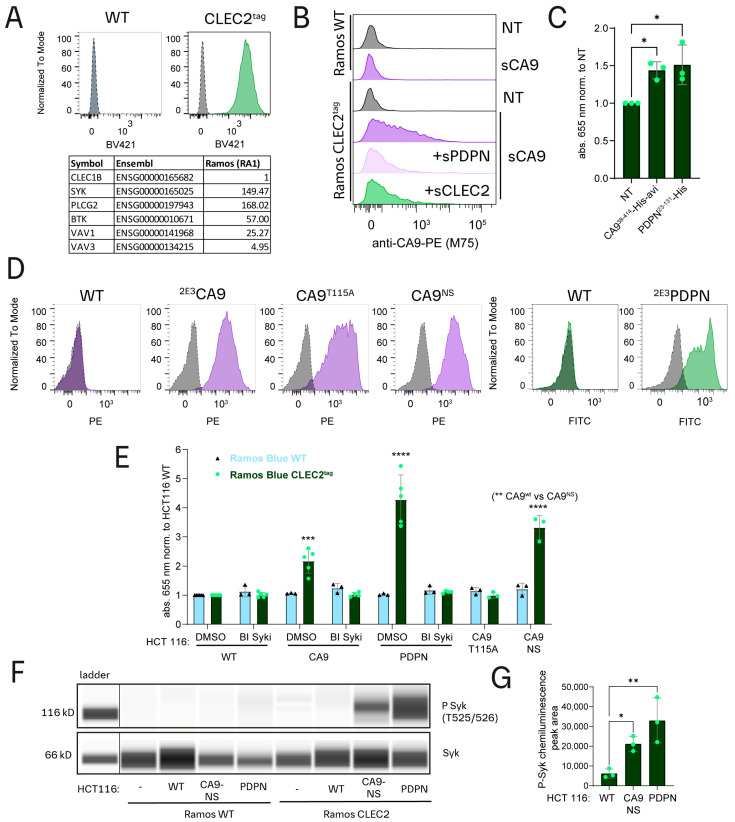
CA9 can induce Syk-dependent signaling CLEC2 pathway. (**A**) Left: CLEC2 cell surface expression was tested on wild-type (WT) Ramos-Blue™ reporter cells (dashed grey line: isotype control; blue solid line anti-CLEC2 test antibody). Right: Stable integration of CLEC2 into Ramos-Blue™ cells is confirmed by flow cytometry (dashed grey line: isotype control; green solid line: anti-CLEC2 test antibody). Bottom: The table shows normalized gene expression (transcript per million (TPM)) values sourced from the ordino database [35] for CLEC2 and selected genes expected to be involved for CLEC2 signaling. (**B**) CA9-His binding tests were conducted on Ramos-Blue™ cells. Binding of soluble CA9 (sCA9) was tested using anti-CA9 (M75) antibody staining. No binding was detected to WT cells, but binding was observed to CLEC2^2E3^ overexpressing Ramos-Blue™ cells. Binding was competed in the presence of soluble PDPN (sPDPN) or soluble CLEC2 (sCLEC2). (**C**) CLEC2^2E3^ overexpressing Ramos-Blue™ were stimulated for 16 h at 37 °C with indicated proteins in absence of FBS. NF-κB controlled expression of reporter gene (secreted embryonic alkaline phosphatase, SEAP) was determined using QUANTI-Blue™. Increased SEAP activity is observed for cells incubated with CA9 and PDPN compared to non-treated (NT) control. One-way ANOVA Dunnett’s multiple comparison test, CA9* = 0.0314, PDPN* = 0.0163. (**D**) Left: Expression tests of full-length CA9 or mutated versions of CA9 in HCT 116 cells were conducted using flow cytometry (dashed grey line: isotype control; purple solid line: anti-CA9 test antibody (M75)). Cell surface expression of CA9 was not detected for HCT 116 cell under normoxia conditions. Stable integration and high cell surface expression of ^2E3^CA9, CA9-T115A or CA9 non-shed version (NS; CA9^Δ393−402^) was detected in engineered HCT 116 cell lines. Right: PDPN expression was not detected on the cell surface of HCT 116 cells but was observed in engineered ^2E3^PDPN overexpressing HCT 116 cells (dashed grey line: isotype control; green solid line: anti-PDPN test staining). (**E**) WT or CLEC2^2E3^ overexpressing Ramos-Blue™ cells were added to fully confluent adherent HCT 116 cells and incubated for 16 h at 37 °C in the absence of FBS. Where indicated, spleen tyrosine kinase Syk inhibitor (0.5 µM BI-1002494 Syk) or DMSO was added. NF-κB activation was determined using QUANTI-Blue™. Activation was detected for Ramos-Blue™ expressing CLEC2 but not WT stimulated with CA9, PDPN or CA9^NS^ expressing HCT 116 cells. Activation was lost in the presence of BI-1002494 and was not detected in T115A mutated CA9. Multiple comparison analysis was performed for WT and CLEC2 Ramos-Blue™ cells individually. One-way ANOVA Šídák’s multiple comparisons test, *** = 0.0006, **** < 0.0001, ** = 0.0035. (**F**) Phosphorylation of Syk protein was observed in Ramos-Blue™ CLEC2^2E3^-expressing cells incubated with HCT 116 cells overexpressing CA9. WT or CLEC2^2E3^ overexpressing Ramos-Blue™ cells were stimulated with HCT 116 cells overexpressing CA9 for 15 min at 37 °C. Reaction was stopped by adding ice-cold cell lysis buffer. The phosphorylation status of Syk protein was then assessed using simple protein immunoassay. An increase in Phospho-Syk signal were found only for CLEC2^2E3^ expressing Ramos-Blue™ cells but not for WT cells when stimulated with HCT 116 expressing CA9^NS^ or ^2E3^PDPN. (**G**) Phospho-Syk signal quantification. Phospho-Syk peak areas were determined using protein simple compass software for simple Western. One-way ANOVA with Šídák’s multiple comparisons test * = 0.04, ** 0.069.

**Figure 6 cells-13-02083-f006:**
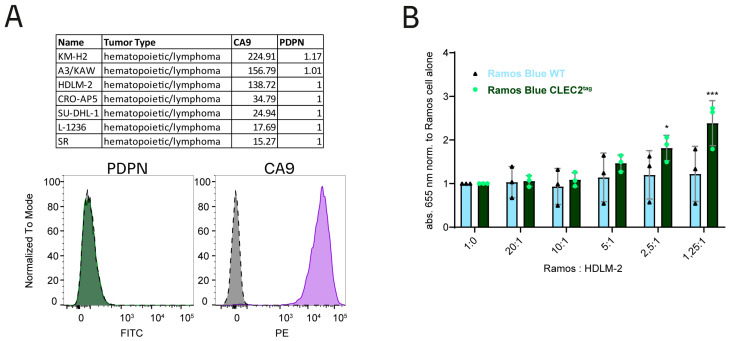
Hodgkin lymphoma (HDLM-2) expressing CA9 but not PDPN induces CLEC2 signaling. (**A**) top: Table shows normalized gene expression (transcript per million (TPM)) values of CA9 or PDPN for hematopoietic cancer cell lines (source: ordino [35]). Bottom: Flow cytometric analysis of CA9 and PDPN in HDLM-2 cells (dashed grey lines: respective isotype control; green solid line: anti-PDPN staining; and purple solid line: anti-CA9 staining). (**B**) WT or CLEC2^2E3^ overexpressing Ramos-Blue™ were mixed with HDLM-2 cells at different ratios and incubated for 16 h at 37 °C in absence of FBS. NF-κB activation was determined using QUANTI-Blue™. Activation was detected for Ramos-Blue™ overexpressing CLEC2 but not for WT cells with a ratio of 2.5:1 and 1.25:1 Ramos-Blue™ to HDLM-2. Multiple comparison analysis was performed for WT and CLEC2^2E3^ Ramos-Blue™ cell lines individually. One-way ANOVA Dunnett’s multiple comparison test, * = 0.0113, *** = 0.0002.

## Data Availability

The original contributions presented in this study are included in the article/Appendix A. Further inquiries can be directed to the corresponding author(s).

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
