# Peer review of "Discovery of Carbonic Anhydrase 9 as a Novel CLEC2 Ligand in a Cellular Interactome Screen"

_cells, 2024, doi:10.3390/cells13242083_

Round 1

Reviewer 1 Report

Comments and Suggestions for Authors

The manuscript by Hoffmann et al. is focused on the development of a new and potent screening approach based on the use of a comprehensive recombinant library of the extracellular domains of human transmembrane proteins. The results obtained using this library indicate that carbonic anhydrase IX (CA9) is a specific ligand of the CLEC-2 receptor. This binding may be very significant, since the CA9/CLEC-2 interaction stimulates Syk, a protein tyrosine kinase functionally linked to CLEC-2.

A few issues may be clarified/resolved to improve the presented study.

1. It is proposed that both podoplanin and CA9 bind to CLEC-2 is due to the presence of the EDLPT fragment in their sequence (line 328). Are there any known membrane protein sporting this or highly similar sequences?

2. Binding of CA9 to CLEC-2 is thought to be mediated by the carbohydrate moiety attached to Thr115 in a fashion similar to podoplanin binding. Mutations in the amino acid sequences are used to prove this point. Does the hydrolysis of carbohydrate parts block the CA9/CLEC-2 interaction?

3. HDLM-2 cells express a very high level of CA9 (Fig. 6) as compared to HCT 116 cells (Fig. 5), but does not seem to induce a stronger stimulation. Why?

4. It would be very useful to see the discussion on the biological significance of the CA9/CLEC-2 interaction outside the cancer context. The CLEC-2-mediated effect of podoplanin seems to be significant for normal individual development; what could be the effect of CA9?

Author Response

The authors would like to thank the reviewer for the positive overall evaluation of the presented work.

Comments 1: “It is proposed that both podoplanin and CA9 bind to CLEC-2 is due to the presence of the EDLPT fragment in their sequence (line 328). Are there any known membrane protein sporting this or highly similar sequences?”

Haji et al. (2022) previously described the binding of human Dectin-1 with CLEC2 via an EDxxT motif. We have incorporated this reference into our manuscript and made the necessary revisions to our text (page 2, lines 57-60 and page 11, lines 342, 349-351). We appreciate the reviewer's important comment, which raised our awareness of the Dectin-1-CLEC2 interaction, and we apologize for having overlooked this report. Overall, the identification of CA9 binding via its EDLPT sequence underscores the importance of the EDxxT motif for CLEC2 interaction. Interestingly, unlike Dectin-1, the EDxxT motif in CA9 is conserved in numerous mammals, not just apes.     

Comments 2: “Binding of CA9 to CLEC-2 is thought to be mediated by the carbohydrate moiety attached to Thr115 in a fashion similar to podoplanin binding. Mutations in the amino acid sequences are used to prove this point. Does the hydrolysis of carbohydrate parts block the CA9/CLEC-2 interaction?”

As pointed out in comment 1, CA9, PDPN and Dectin-1 share a EDxxT CLEC2 binding motif. The dependency of the carbohydrate modification for this motif has been shown for PDPN and Dectin-1. Glycosylation of Thr115 of CA9’s EDxxT motif has been described previously. Overall, this very strongly indicate a similar importance of glycosylation for the CA9-CLEC2 interaction. We tested if our CLEC2 library construct would bind to non-treated or neuraminidase treated HCT 116 cells expressing CA9 or PDPN cells. Our results further confirm that the sialylation of the EDxxT motif is important for CLEC2 binding (Figure 4E and page 11, lines 363-368). We thank the reviewer for this suggestion.

Comments 3: “HDLM-2 cells express a very high level of CA9 (Fig. 6) as compared to HCT 116 cells (Fig. 5), but does not seem to induce a stronger stimulation. Why?”

In contrast to HCT 116 cells, HDLM2 cells are not adherent but suspension cells. This is why we think it may be difficult to compare the two cell lines in their ability to activate Ramos reporter cell line.

Comments 4: “It would be very useful to see the discussion on the biological significance of the CA9/CLEC-2 interaction outside the cancer context. The CLEC-2-mediated effect of podoplanin seems to be significant for normal individual development; what could be the effect of CA9?”

We agree with the reviewer that these are a very interesting question, and we are looking forward to learning more about the CA9-CLEC2 relevance by future investigations.

We expanded our discussion and commented on CA9 potential role outside of the cancer context (pages 17-18, lines 588-599).

Reviewer 2 Report

Comments and Suggestions for Authors

Hoffmann et al, critically examined the interactions of membrane proteins with immune cells, focusing on their potential as therapeutic targets. The authors employed a recombinant protein library of human extracellular domains and ER-Golgi-lysosomal system proteins and used for flow cytometric screening to identify binding events. The authors identified CLEC2 as a robust CA9 binding partner.  Like PDPN, CA9 activated CLEC2 signaling via Syk, with evidence from a Hodgkin’s lymphoma cell line demonstrating Syk-dependent CLEC2 activation by CA9. This study validates CA9 as an endogenous CLEC2 ligand and uncovers novel CA9 functions in hematological cancers.

The authors tried to support their hypothesis by using a series of genetic models, and the study is well-prepared and meticulous.  Overall, this work expands the understanding of CA9 interactions, identifying potential therapeutic targets and paving the way for further exploration in cancer biology and immune cell modulation and should be published in Cells, provided minor issues related to the primary observations are addressed and confirmed.

Minor comments;

1. Figure 1D & Supplemental table S5

From the protein-to-cell flow cytometry screen, the authors set a cutoff value of 1.5× greater mean fluorescence intensity (MFI) for recombinant library proteins binding to CD14-high stained cells. However, the rationale for selecting this specific cutoff value is unclear. 

Additionally, the screening method used does not exclude hits resulting from indirect binding. Clarifying the rationale for the cutoff value and addressing the limitations of indirect binding in the main text would provide better context and enhance the reader's understanding of the methodology.

2. Page 8. Figure 2A 

The description of the statistical method is missing. 

3. Page 10. Figure 4B

Conducting a binding competition test at a single concentration point is not considered best practice. The authors should include additional concentration points between 0 and 0.5 to demonstrate whether the binding occurs in a dose-dependent manner. Additionally, the figure labeling is unclear and would benefit from revision to make it more intuitive for readers.

4. Page 6 Line 221 

repeated period mark. 

Author Response

We appreciate that the reviewer considers this work as an important contribution for future exploration in cancer biology and would like to thank the reviewer for the positive feedback and the important comments addressed below.

Comments 1: “Figure 1D & Supplemental table S5

From the protein-to-cell flow cytometry screen, the authors set a cutoff value of 1.5× greater mean fluorescence intensity (MFI) for recombinant library proteins binding to CD14-high stained cells. However, the rationale for selecting this specific cutoff value is unclear. 

Additionally, the screening method used does not exclude hits resulting from indirect binding. Clarifying the rationale for the cutoff value and addressing the limitations of indirect binding in the main text would provide better context and enhance the reader's understanding of the methodology.”

Response 1: We thank the reviewer for pointing this out. We added our rationale for the selection in the main text (page 5, lines 197-203).

We agree with the assessment of the reviewer and expanded our discussion including the possibility for indirect binding in our screen. Moreover, we also added other limitations of the presented screen (page 16, lines 516-528).

Comments 2: “Page 8. Figure 2A 

The description of the statistical method is missing.“

Response 2: Thank you for pointing this out. We added the missing information (page 8, lines 286-287).

Comments 3: ”Page 10. Figure 4B

Conducting a binding competition test at a single concentration point is not considered best practice. The authors should include additional concentration points between 0 and 0.5 to demonstrate whether the binding occurs in a dose-dependent manner. Additionally, the figure labeling is unclear and would benefit from revision to make it more intuitive for readers.”

Response 3: We repeated the experiment with buffy-coat derived platelets using additional PDPN concentrations and improved the labeling of the figure. We observe a dose-dependent competition for CA9 binding to CD42b stained platelets in presence of PDPN (Figure 4B).

Comments 4: “Page 6 Line 221 

repeated period mark.” 

Response 4: The additional period mark is removed.

Reviewer 3 Report

Comments and Suggestions for Authors

Dear Authors,

The manuscript entitled "Discovery of Carbonic Anhydrase 9 as a novel CLEC2 ligand in a cellular interactome screen" is a very well performed and interesting study in the filed revealing the potential role of CA9. Indeed, the whole study is well conducted and in my opinion the performed experiments support great the discussion. Only minor revisions, i have to report for the submitted study.

1) Did the authors evaluate the interaction of CA9 besides monocytes with other immune cells. It would be great to provide information regarding the potential interaction with the T cells.

2) Also, it would be very good for the whole manuscript, the authors to include immunofluoresence experiments to show the interactions between platelets and monocytes.

3) In materials and methods, where flow cytometry has been performed (2.4. Protein-to-cell interaction and 2.5. Protein-to-cell binding experiments) please provide further details, regarding the gating strategy, and events counted to obtain the presented results.

Author Response

The authors would like to thank the reviewer for the positive overall evaluation of the presented work.

Comments 1: “Did the authors evaluate the interaction of CA9 besides monocytes with other immune cells. It would be great to provide information regarding the potential interaction with the T cells.”

Response 1: We tested binding of CA9 to PBMCs (Figure S1C). We did not find significant binding of CA9 to T cells, B cells or NK cells indicating either no CA9 binding to other peripheral blood cells or binding below our detection limit.

Comments 2: “Also, it would be very good for the whole manuscript, the authors to include immunofluoresence experiments to show the interactions between platelets and monocytes.”

Response 2: While we agree with the reviewer that a more visual representation of a monocyte/platelet interaction could be helpful for a clear understanding of our manuscript, we are concerned that these images may not be very representative/conclusive. Platelets are very abundant in blood and would likely also by chance be found close to or on top of a monocyte making it difficult to conclude if there is a direct interaction by imaging. We believe that our co-staining of CD14 and CD41 presented in figure 2B and S2C presents an alternative solution to confirm this interaction in our screening set-up.

Comments 3: “In materials and methods, where flow cytometry has been performed (2.4. Protein-to-cell interaction and 2.5. Protein-to-cell binding experiments) please provide further details, regarding the gating strategy, and events counted to obtain the presented results.”

Response 3: We provided additional information (page 3, lines 108-109, 126-128). As pointed out by reviewer 2, we also added a more detailed discussion regarding the limitations of our flow cytometric screen. The binding identified here are likely biased towards the dominant CD14high population due to underrepresentation of certain subtypes but also due to varying protein concentrations present in the conditioned media used for the screening.